# Flood induced phenotypic plasticity in amphibious genus *Elatine* (Elatinaceae)

Attila Molnár V.[1,*], János Pál Tóth[2,*], Gábor Sramkó[1,3], Orsolya Horváth[1], Agnieszka Popiela[4], Attila Mesterházy[5] and Balázs András Lukács[6]

[1] Department of Botany, University of Debrecen, Debrecen, Hungary
[2] MTA-DE, "Lendület" Behavioural Ecology Research Group, University of Debrecen, Debrecen, Hungary
[3] MTA-ELTE-MTM Ecology Research Group, Budapest, Hungary
[4] Department of Botany and Nature Conservation, University of Szczecin, Szczecin, Poland
[5] Department of Botany, University of West-Hungary, Sopron, Hungary
[6] Department of Tisza Research, MTA Centre for Ecological Research, Debrecen, Hungary
[*] These authors contributed equally to this work.

## ABSTRACT

Vegetative characters are widely used in the taxonomy of the amphibious genus *Elatine* L. However, these usually show great variation not just between species but between their aquatic and terrestrial forms. In the present study we examine the variation of seed and vegetative characters in nine *Elatine* species (*E. brachysperma, E. californica, E. gussonei, E. hexandra, E. hungarica, E. hydropiper, E. macropoda, E. orthosperma* and *E. triandra*) to reveal the extension of plasticity induced by the amphibious environment, and to test character reliability for species identification. Cultivated plant clones were kept under controlled conditions exposed to either aquatic or terrestrial environmental conditions. Six vegetative characters (length of stem, length of internodium, length of lamina, width of lamina, length of petioles, length of pedicel) and four seed characters (curvature, number of pits / lateral row, 1st and 2nd dimension) were measured on 50 fruiting stems of the aquatic and on 50 stems of the terrestrial form of the same clone. MDA, NPMANOVA Random Forest classification and cluster analysis were used to unravel the morphological differences between aquatic and terrestrial forms. The results of MDA cross-validated and Random Forest classification clearly indicated that only seed traits are stable within species (i.e., different forms of the same species keep similar morphology). Consequently, only seed morphology is valuable for taxonomic purposes since vegetative traits are highly influenced by environmental factors.

Corresponding author
Attila Molnár V.,
mva@science.unideb.hu

## INTRODUCTION

Environmentally induced phenotypic change plays a key role in the adaptation of organisms to rapidly changing environmental conditions (*Bradshaw, 1965*; *Schlichting, 1986*). This phenomenon is especially important for aquatic and semi aquatic plants (*Wells & Pigliucci, 2000*; *Kaplan, 2002*; *Dorken & Barrett, 2004*) which enables them to survive and

reproduce in heterogeneous and temporarily highly variable environments. Water depth is a temporally and spatially changing dynamic factor in wetlands and littoral communities (*Rea & Ganf, 1994*). Although the morphological (*Nielsen & Sand-Jensen, 1997*), ecological (*Volder, Bonis & Grillas, 1997*; *Warwick & Brock, 2003*; *Lin, Alpert & Yu, 2012*), and physiological (*Valanne, Aro & Rintamäki, 1982*; *Laan & Blom, 1990*; *Robe & Griffiths, 1998*; *Mommer & Visser, 2005*; *Klančnik, Mlina & Gaberščik, 2012*) aspects of phenotypic plasticity are well studied among the aquatics, its importance has been underestimated in taxonomical and evolution studies on plants (*Davis & Heywood, 1963*; *Kaplan, 2002*).

Phenotypic plasticity maximises plant fitness in a variable environment and (*Bradshaw, 1965*; *Wright & McConnaughay, 2002*), thus, can play an important role in adaptation to amphibious environments. When cultivated under moist conditions, many of the freshwater angiosperms can be induced to transform into small terrestrial forms. It has been recorded that this phenomenon sometimes appears in certain cases of aquatic species like *Nymphaea alba, Nuphar lutea, Myriophyllum* and *Utricularia* spp. In nature, the production of terrestrial form from these aquatic species can greatly contribute to their survival over periods of temporal drought in less humid areas (*Hejný, 1960*; *Den Hartog & Segal, 1964*).

Amphibious aquatics are adapted to a dual-life; under submerged conditions they develop aquatic forms, whereas the same individual can have a different terrestrial form in open air. This duality in life history can involve surprising physiological alterations (*Ueno et al., 1988*; *Ueno, 1998*; *Agarie et al., 2002*); all of the amphibious species have the ability to photosynthesize on air by developing air leaves or terrestrial shoots (*Maberly & Spence, 1989*). Hence these species are exposed to extreme conditions of temperature, availability of gases and solar radiation (*Germ et al., 2002*). They usually live in the littoral zone of lakes, wetlands and rivers or ephemeral wetlands, where their phenotypic plasticity is a key factor for survival in their temporal and fast changing environment (*Deil, 2005*).

Several genera of aquatic plants have amphibious habits but it is rare for a whole genus to be adapted to live in temporal waters. The genus *Elatine* contains ca. 15–25 ephemeral, amphibious species (*Heywood et al., 2007*) that are widespread in the temperate regions of both hemispheres. Surprisingly, there is only a few studies dealing with the causal relationship between their morphology and environmental variables and its effect on their taxonomy (*Popiela & Łysko, 2010*; *Popiela et al., 2011*; *Popiela et al., 2012*)—a telling fact is that the last worldwide monograph on *Elatine* was published more than 140 years ago (*Dumortier, 1872*). Amongst the main causes of this obscurity are probably their enigmatic rarity, erratic temporal appearance that depends mainly on environmental factors like the amount of precipitation and the extent of inundation (*Takács et al., 2013*). Unquestionably, the high degree of the morphological variability of *Elatine* also contributes to the taxonomic uncertainties, which is evidently connected to their amphibious life-history. The clonal nature of *Elatine* also contributes to their morphological variability, because large clonal plants are especially exposed to variation in water depth over time and space (*Vretare et al., 2001*).

The main distinguishing characteristics of *Elatine* species are related to flower and seed morphology (*Cook, 1968a*; *Brinkkemper et al., 2008*; *Uotila, 2009*; *Uotila, 2010*; *Molnár et al., 2013*; *Molnár, Popiela & Lukács, 2013*), but vegetative traits (i.e., relative length of pedicel, sepals or petals, form of leaves, etc.) are also frequently used in descriptions of *Elatine* taxa (*Wight, 1831*; *Albrecht, 2002*; *Lægaard, 2008*). An example is the length of pedicel, which has great importance in separation of some species-pairs (e.g., *E. ambigua* and *E. triandra; E. hungarica* and *E. campylosperma; E. gussonei* and *E. hydropiper*), but the taxonomic value of such characters are highly questionable. Even though the unusual degree of morphological variability and the crucial importance of *in vitro* cultural studies in the genus were pointed out more than 60 years ago by *Mason* (*1956*: 239): '*The differences between aquatic and terrestrial forms of the same species often seem greater than the differences between species*' and '*The genus is in need of a thorough cultural study designed to test the nature of characters and their validity as criteria of species*'. According to the best of our knowledge, such experiments have not been accomplished and published yet.

As part of our ongoing researches aiming at the taxonomic clarification of the genus *Elatine* in Europe, we examine the level of phenotypic plasticity in the genus in order to lay down the basis of a comprehensive taxonomic study. More specifically, we provide here a study of seed and vegetative traits concerning the aquatic and terrestrial form of nine *Elatine* species studied in a laboratory culture system. Our aims were to (i) quantify the degree of phenotypic plasticity in case of vegetative organs and seeds, and (ii) to examine the phenotypic overlap among the species, and then (iii) determine which type of traits could be used to differentiate the species in practical identification. This is done in hope of serving as a base for future taxonomic works in the genus *Elatine*, including a practical guide to the thoughtful usage of morphological variation in this genus.

## MATERIAL AND METHODS

### Plant material and cultivation

We set up a cultivation experiment to study the plastic variation of *Elatine* species in waterlogged and submerged conditions. To eliminate the effect of genetic variation within the studied species we used only clones of the same individual for any comparison of morphological differences. Seeds of nine annual, clonal *Elatine* species collected from indigenous populations were included for the present study (Table 1). *Elatine hungarica*, *E. hydropiper* and *E. triandra* are protected species and were sampled in Hungary with the permission of the Hortobágy National Park Directorate (Permission id.: 45-2/2000, 250-2/2001). We only collected seeds from one aquatic form specimen of all Elatine species because its submerged condition makes it autogamous and ensures that different capsules contain seeds with the same genetic information. Seeds were sown in 12.5 × 8.5 cm plastic boxes on sterilised (autoclaved for 3 h, 180 °C) soil, which was continuously wetted and germinated in the laboratory of the Department of Botany at University of Debrecen. Plantlets were grown in climate controlled rooms (with 14 h/day light and 30 μmol $m^{-2}$ $s^{-1}$ light intensity, temperatures: under light 22 ± 2 °C and under darkness 18 ± 2 °C). Two specimens of one week old plantlets from each species were transplanted,

**Table 1 Taxonomic position, distribution and sample source of nine *Elatine* species studied.**

| Species | Section | Distribution | Locality |
|---|---|---|---|
| *Elatine brachysperma* A. Gray | *Triandra* Seubert | North-American (*The PLANTS Database, 2014*) | USA: Fallbrook (CA) (N33.4°, W117.4°) |
| *Elatine californica* A. Gray | *Elatinella* Seubert | North-American (*The PLANTS Database, 2014*) | USA: Los Angeles (CA) (N33.8°, W118.3°) |
| *Elatine gussonei* (Sommier) Brullo et al. | *Elatinella* Seubert | Central Mediterranean (*Molnár, Popiela & Lukács, 2013*) | Sicily: Modica (N36°, E14.7°) |
| *Elatine hexandra* (Lapierre) DC. | *Elatinella* Seubert | Sub-Atlantic and Central-European (*Popiela et al., 2011*) | Poland: Poznan (N51.55°, E17.35°) |
| *Elatine hungarica* Moesz | *Elatinella* Seubert | Temperate Eurasian (*Takács et al., 2013*) | Hungary: Konyár (N47.3°, E21.7°) |
| *Elatine hydropiper* L. | *Elatinella* Seubert | Euro-Siberian (*Popiela et al., 2012*) | Hungary: Tiszagyenda (N47.4°, E20.5°) |
| *Elatine macropoda* Guss. | *Elatinella* Seubert | Mediterranean (*Popiela & Łysko, 2010*) | Sardinia: Olmedo (N40.6°, E8.4°) |
| *Elatine orthosperma* Düben | *Elatinella* Seubert | Northern European (*Uotila, 1978*) | Finland: Oulu (N65.0°, E25.4°) |
| *Elatine triandra* Schkuhr | *Triandra* Seubert | Cosmopolitan (*Popiela et al., 2015*) | Hungary: Kisköre (N47.5°, E20.5°) |

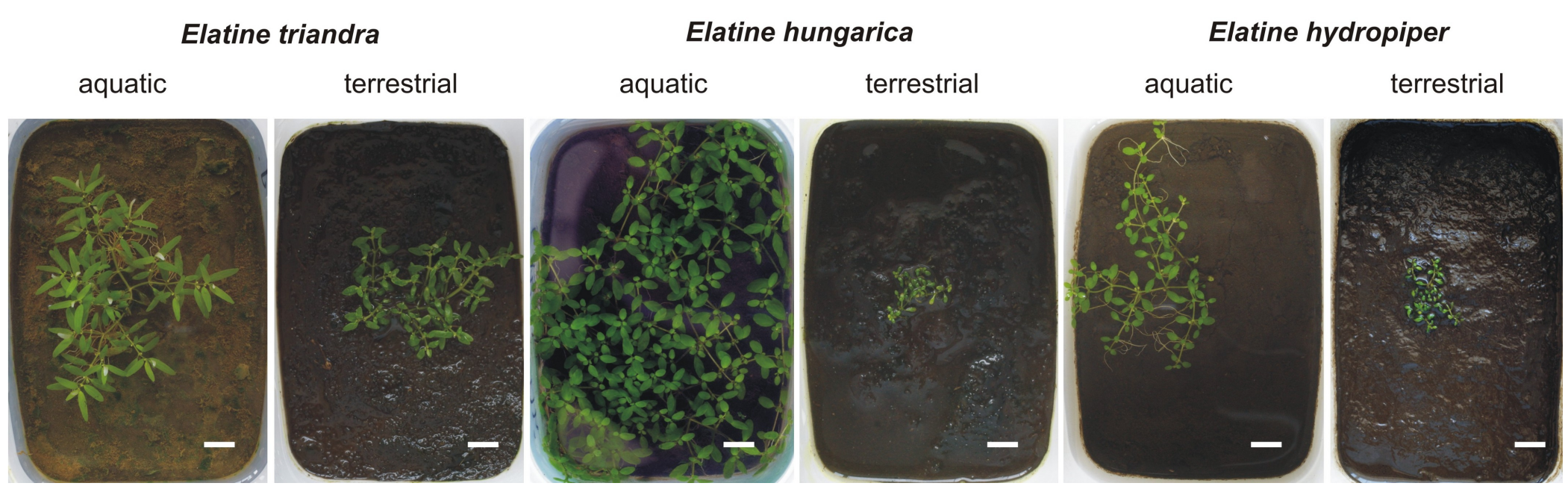

**Figure 1 Aquatic (continuously flooded) and terrestrial (growing on wet mud) forms with same age of three central European *Elatine* species cultivated in plastic boxes.** Scale bars represent 10 mm.

then one specimen was grown under continuous water cover to develop into aquatic form, while another one (terrestrial forms) was grown on wet mud until they both reached the fruiting stage and formed a clone bed with minimum 50 fruiting stems, between 45 and 70 days (Fig. 1). For the morphological study six traits (length of stem, length of internode, length of lamina, width of lamina, length of petioles, length of pedicel) were measured on 50 fruiting stems of the aquatic and on 50 stems of the terrestrial form of the same clone using calliper (0.1 mm accuracy). Leaf traits and internodes were measured on 3 leaves of each specimens. 3 capsules were collected from each sample. Then seeds were pooled and 50 randomly collected seeds were photographed from each clone and four traits (curvature (°), number of pits/lateral row, 1st dimension (mm), 2nd dimension (mm)) were measured on digital images (Fig. 2). Curvature of seeds was measured following the method of *Mifsud (2006)*.

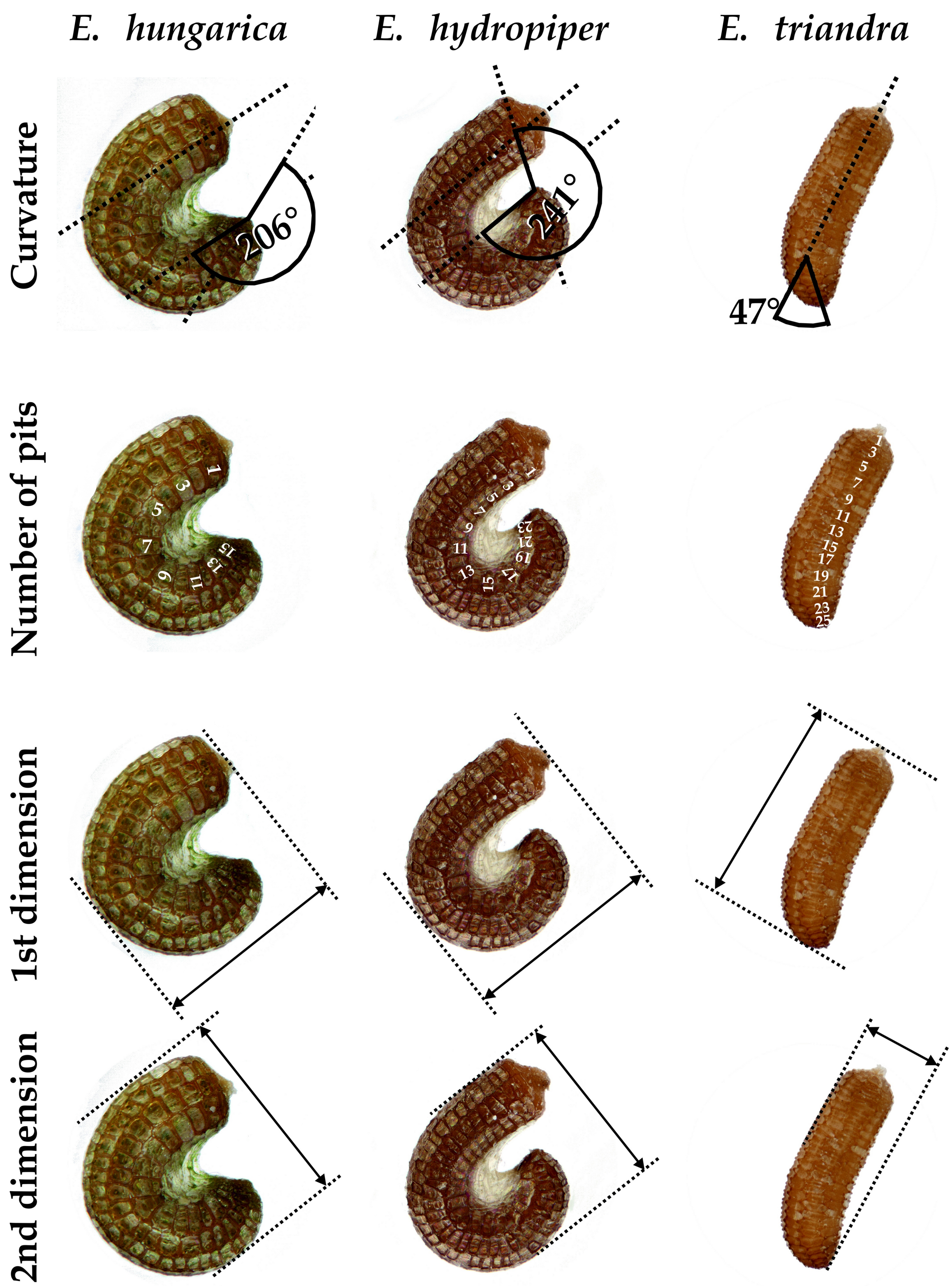

Figure 2 **Seed traits measured as examplified by three *Elatine* species studied.**

### Data analyses

Multivariate and univariate statistical analyses were applied to determine the validity of vegetative and seed traits. Multiple Discriminant Analysis (Linear Discriminant Analysis for more than two groups) was used to reveal morphological differences between terrestrial and aquatic forms based on vegetative and seed traits using SPSS 16.

In the analyses the predefined groups were the two ecological forms of the studied species. Mean scores of our predefined groups were plotted to illustrate the pattern of morphological differentiation. Wilks's $\lambda$ was used to measure the discriminatory power of the model. Its values change from 0 (perfect discrimination) to 1 (no discrimination). For visualise the relationship between the different species and forms based on vegetative and seed characters Mahalanobis distance based UPGMA trees were constructed.

To test the statistical significance of the visible pattern obtained by MDA and UPGMA trees, we used Mahalanobis distance based Permutational Multivariate Analysis of Variance (NPMANOVA), since some of our variables do not show normal distribution. The number of permutations was set to 10,000. Linear discriminant analysis frequently achieves good performances in the tasks of face and object recognition, even though the assumptions of common covariance matrix among groups and normality are often violated (*Duda, Hart & Stork, 2001*; *Li, Zhu & Ogihara, 2006*).

Classification of our groups was made using the cross-validated grouping function in SPSS. In this method, one known specimen is left out at a time, and assigned using the discriminant function which is calculated based on all the cases except the given case. The numbers of correct assignments were used to evaluate the usefulness of the discriminant function. High numbers of correct assignments indicate diagnostic differences between the surveyed groups.

Random Forest was also used to determine variable importance and classification accuracy in vegetative and seed characters (*Liaw & Wiener, 2002*). Random Forest is an algorithm (*Breiman, 2001*) for classification that uses an ensemble of classification trees. Each of the classification trees is built using a bootstrap sample of the data, and at each split the candidate set of variables is a random subset of the variables. The results of MDA and Random Forest classification have been presented as a confusion matrix.

The most discriminative traits were also tested independently by the non-parametric Kruskal–Wallis test using R computing environment (*R Core Team, 2014*). The results are interpreted by the kruskalmc function in pgirmess package. kruskalmc makes multiple comparisons of treatments.

## RESULTS

### Vegetative traits

The vegetative traits of the aquatic or terrestrial forms of the nine *Elatine* species were different with high discriminatory power (Wilks's $\lambda = 0.0001$, $p < 0.001$). The first two axes explained 67% of variance (43% of axis 1 and 24% of axis 2, respectively). The length of the 3rd lamina (−0.593), length of the 1st lamina (−0.591), length of stem (0.505), and length of the 2nd lamina (−0.477) had the highest relative importance in the first

function based on the standardized canonical discriminant function coefficients values. In the second function the most important variables are length of stem (0.401), length of the 2nd lamina (0.782), and width of the 1st lamina (−0.823). The scatter plot of group mean scores on the first two canonical axes showed a relatively large distance between the aquatic and the terrestrial forms of the same species (Fig. 3B). These distances are sometimes greater than the distance between the different species (Fig. 3D). The cross-validated classification correctly assigned 77.7% of the specimens. The lowest assignment success was in case of *E. hexandra* (aquatic) (38%) and *E. hungarica* (terrestrial) (30%) (see: Table 2). The Random Forest variable importance analysis also indicate the importance of the length of pedicel, the 1st lamina, the stem and the 1st petiole (Fig. 4). The success rate of Random Forest classification was 82.33% (Table 3). The variation of important vegetative traits indicated substantial differences between the terrestrial and aquatic forms within the species, however the variation of each forms has high overlaps between the species (Fig. 5).

The results of the NPMANOVA indicated all predefined groups were significantly different from each other ($p < 0.05$). On the UPGMA tree the different forms of the same species clustered to different branches with the exception of *E. macropoda* and *E. gussonei* (Fig. 3D).

Univariate analysis on the measured vegetative traits indicated significant differences between the different ecological forms of the same species. None of the vegetative traits were alone suitable for species identification (see Table 4, Figs. 3B and 3D).

## Seed traits

The seed traits of the aquatic or terrestrial forms of the nine *Elatine* species differed significantly (Wilks's $\lambda = 0.001$, $p < 0.001$). The first two axes explained more than 83% of the total variance between groups (52% of axis 1 and 31% of axis 2, respectively). Curvature (0.873) and the 2nd dimension (0.47) showed the largest loadings in the first discriminant function based on the standardized canonical discriminant coefficient values, while in the second discriminant function the number of pits on the testa in a lateral row (0.832) and the 1st dimension (0.62) had notable loadings. The group centroids of the aquatic and terrestrial forms of the same species are positioned very close to each other, and at the same time the species are well separated with the exception of *Elatine hungarica* and *E. californica* (Fig. 3C).

The cross-validated classification could assign only 50.2% of the specimens correctly to the predefined groups, although the success of assignments at the species level is usually high 83.8% (Table 5). The lowest level of correct assignments occurred between *E. californica* (62%) and *E. hungarica* (57%).

The Random Forest variable importance indicate that the curvature and the number of pits are the most useful characters in classification. (Fig. 6). The success rate of Random Forest classification was 49.78% (Table 6). The average classification success is 87.5% in species level. The within species variation of important seed traits did not differ between the terrestrial and aquatic forms, and the variation of each form had only small overlaps between the species (Fig. 7).

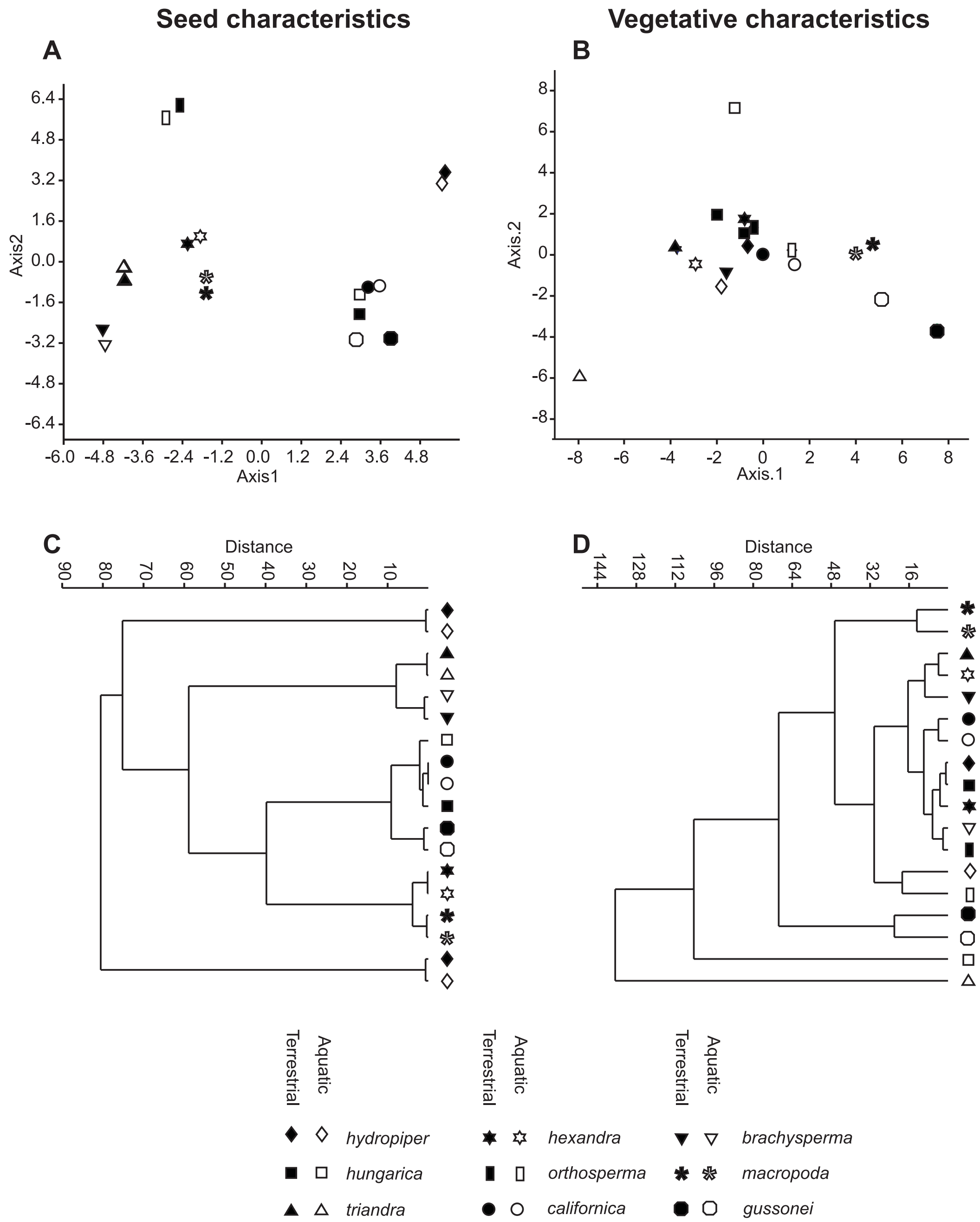

**Figure 3 Morphological relationships among the surveyed *Elatine* species as displayed by MDA scatterplots (A, B) and UPGMA cluster diagrams (C, D).** Symbols indicate the group based on seed traits (A, C) and on vegetative traits (B, D).

The seed trait based NPMANOVA indicated significant differences ($p < 0.05$) between the species but differences between the two ecological forms of the same species were not significant with three exceptions. The two forms of *E. gussonei* ($p = 0.03$) and the aquatic and terrestrial forms of *E. hungarica* and *E. hydropiper* ($p < 0.05$) proved to be different. We also tested the usefulness of the measured seed traits independently. The

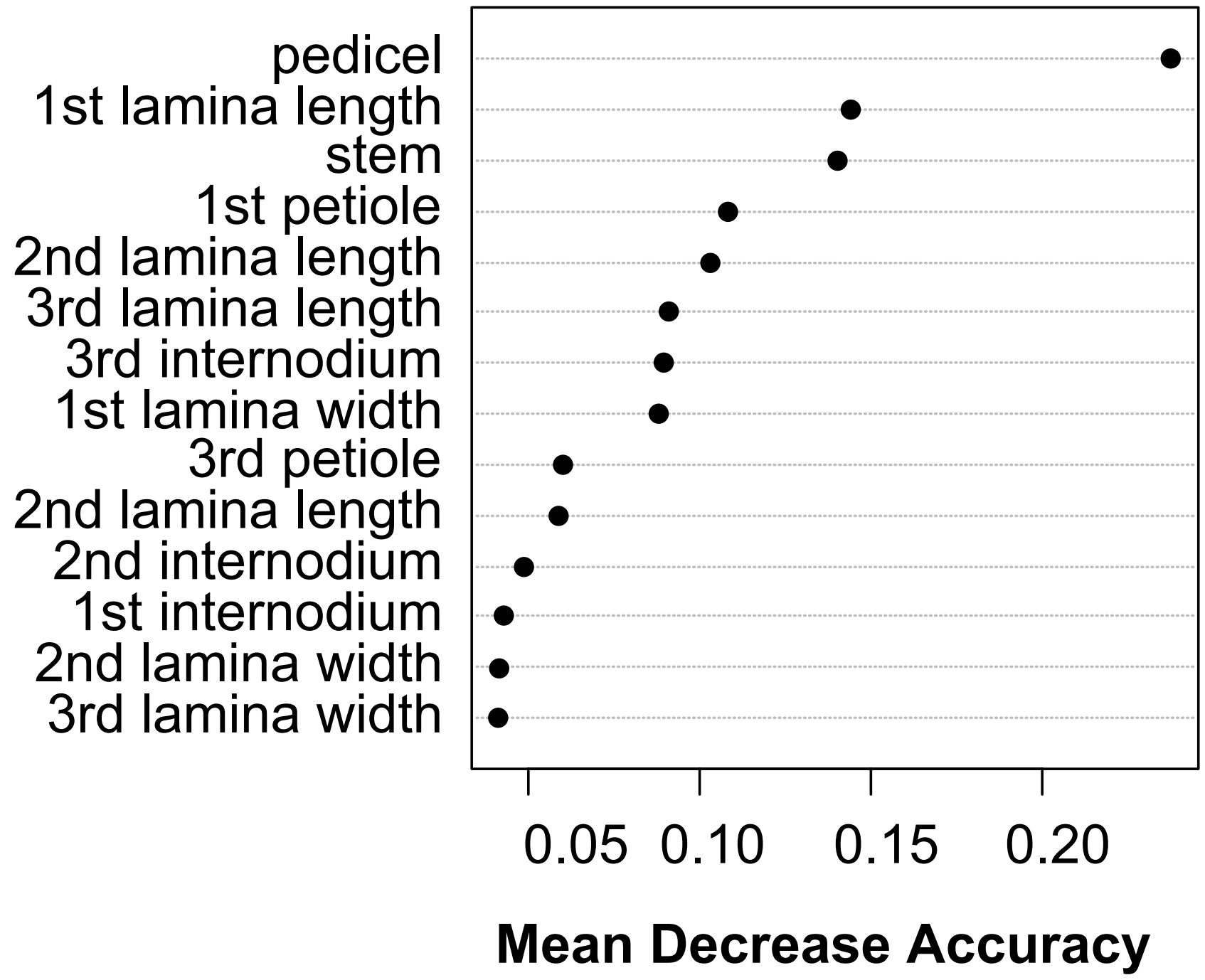

**Figure 4 Dotchart of variable importance as measured by a Random Forest for vegetative traits (*Liaw & Wiener, 2002*).**

Kruskall–Wallis test found none of the seed traits to be suitable for perfect discrimination of all species alone, although different forms of the same species are not significantly separable (Table 7 and Fig. 5).

## DISCUSSION

Phenotypic plasticity is the ability of an organism to change its phenotype in response to relatively rapid changes of its environment (*Price, Qvarnström & Irwin, 2003*). This was documented for several aquatic plants, e.g., *Potamogeton* (*Idestam-Almquist & Kautsky, 1995*; *Kaplan, 2002*) and *Batrachium* (*Cook, 1968b*; *Garbey, Thiébaut & Muller, 2004*; *Garbey, Thiébaut & Muller, 2006*). An important type of potentially adaptive plasticity involves differences in morphological, anatomical and physiological characteristics of leaves along environmental gradients such as light and/or water availability (*Wells & Pigliucci, 2000*). Nonetheless, if distinctive morphological features of taxa depend on environmental conditions, phenotypic plasticity may cause taxonomic errors. Plant taxonomy is sensible of errors when forms of a species are erroneously named as distinct taxa (*Kaplan, 2002*; *Sultan, 2004*). Understanding this issue in a threatened and vulnerable genus such as *Elatine* can contribute to a clarified taxonomy that is essential for an effective conservation. *Mason (1956)* highlighted that the taxonomy of *Elatine* suffers from the high levels of phenotypic plasticity. According to his opinion several *Elatine* species or ecotypes of a species were classified into wrong taxa due to the phenotypic variation displayed. For example *Elatine hungarica*, which is listed on IUCN Red List as data deficient taxon (*Bilz et al., 2011*),was merged to *Elatine hydropiper* (*Cook, 1968a*; *Casper & Krausch, 1980*) based

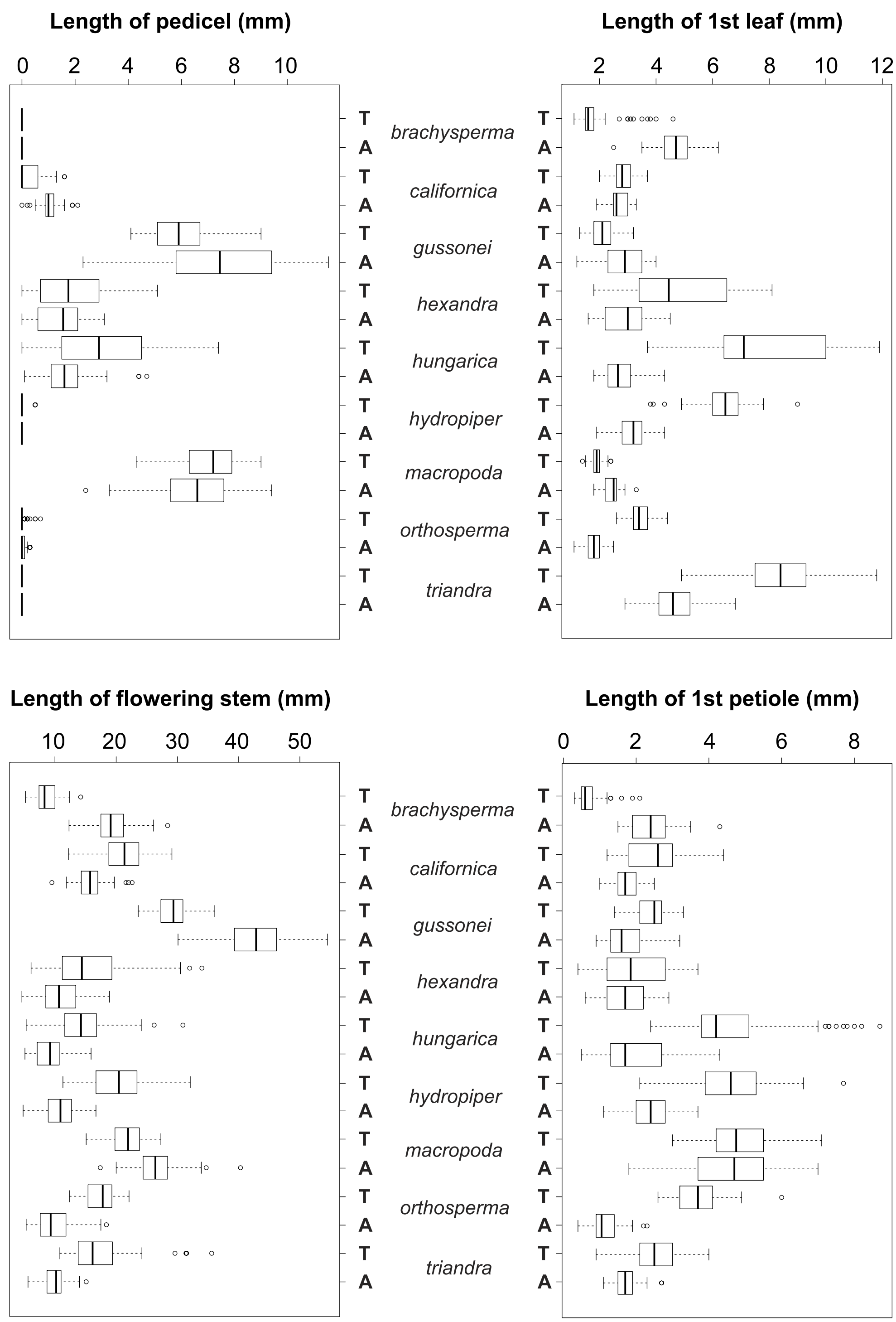

**Figure 5 Boxplots of the most discriminative vegetative traits among the nine *Elatine* species studied.** Terrestrial (T) and aquatic (A) forms significantly differed in all the species; the aquatic forms are relatively larger than terrestrial ones. Notations: Boxes mean 25–75 percentiles, lines are medians, squares are means, whiskers are standard deviations.

**Table 2** **MDA Cross validated classification based on vegetative traits.** Rows: given group; columns: predicted groups. The 77.7% of the specimens are correctly assigned.

| | braAq | braTe | hydTe | hydAq | triTe | triAq | ortTe | ortAq | hexTe | hexAq | macTe | macAq | gusTe | gusAq | calTe | calAq | hunTe | hunAq | Total |
|---|---|---|---|---|---|---|---|---|---|---|---|---|---|---|---|---|---|---|---|
| braAq | 46 | 0 | 0 | 2 | 0 | 0 | 0 | 0 | 0 | 0 | 0 | 0 | 0 | 0 | 1 | 1 | 0 | 0 | 50 |
| braTe | 0 | 40 | 1 | 0 | 0 | 0 | 1 | 0 | 8 | 0 | 0 | 0 | 0 | 0 | 0 | 0 | 0 | 0 | 50 |
| hydTe | 0 | 4 | 28 | 0 | 0 | 0 | 2 | 1 | 3 | 1 | 0 | 0 | 0 | 0 | 3 | 0 | 8 | 0 | 50 |
| hydAq | 5 | 0 | 2 | 41 | 0 | 0 | 0 | 1 | 0 | 1 | 0 | 0 | 0 | 0 | 0 | 0 | 0 | 0 | 50 |
| triTe | 0 | 0 | 0 | 0 | 45 | 0 | 0 | 0 | 2 | 3 | 0 | 0 | 0 | 0 | 0 | 0 | 0 | 0 | 50 |
| triAq | 0 | 0 | 0 | 1 | 0 | 46 | 0 | 0 | 0 | 3 | 0 | 0 | 0 | 0 | 0 | 0 | 0 | 0 | 50 |
| ortTe | 0 | 6 | 0 | 0 | 0 | 0 | 36 | 0 | 2 | 0 | 0 | 0 | 0 | 0 | 0 | 0 | 6 | 0 | 50 |
| ortAq | 0 | 0 | 2 | 0 | 0 | 0 | 0 | 48 | 0 | 0 | 0 | 0 | 0 | 0 | 0 | 0 | 0 | 0 | 50 |
| hexTe | 0 | 9 | 1 | 0 | 0 | 0 | 3 | 1 | 32 | 0 | 0 | 0 | 0 | 0 | 1 | 0 | 3 | 0 | 50 |
| hexAq | 6 | 7 | 1 | 0 | 6 | 0 | 0 | 0 | 6 | 19 | 0 | 0 | 0 | 0 | 3 | 0 | 2 | 0 | 50 |
| macTe | 0 | 0 | 0 | 0 | 0 | 0 | 0 | 1 | 0 | 0 | 42 | 4 | 0 | 0 | 0 | 3 | 0 | 0 | 50 |
| macAq | 0 | 0 | 0 | 0 | 0 | 0 | 0 | 3 | 0 | 0 | 0 | 46 | 0 | 1 | 0 | 0 | 0 | 0 | 50 |
| gusTe | 0 | 0 | 0 | 0 | 0 | 0 | 0 | 0 | 0 | 0 | 0 | 0 | 40 | 9 | 0 | 1 | 0 | 0 | 50 |
| gusAq | 0 | 0 | 0 | 0 | 0 | 0 | 0 | 0 | 0 | 0 | 0 | 0 | 0 | 49 | 0 | 1 | 0 | 0 | 50 |
| calTe | 0 | 0 | 1 | 0 | 0 | 0 | 2 | 0 | 0 | 0 | 0 | 0 | 0 | 0 | 41 | 5 | 1 | 0 | 50 |
| calAq | 0 | 0 | 1 | 0 | 0 | 0 | 0 | 1 | 0 | 0 | 0 | 2 | 0 | 0 | 9 | 37 | 0 | 0 | 50 |
| hunTe | 0 | 6 | 7 | 0 | 0 | 0 | 5 | 5 | 9 | 0 | 0 | 0 | 0 | 0 | 3 | 0 | 15 | 0 | 50 |
| hunAq | 0 | 1 | 0 | 0 | 0 | 0 | 0 | 0 | 0 | 0 | 0 | 0 | 0 | 0 | 1 | 0 | 0 | 48 | 50 |
| Total | 57 | 73 | 44 | 44 | 51 | 46 | 49 | 61 | 62 | 27 | 42 | 52 | 40 | 59 | 62 | 48 | 35 | 48 | 900 |

**Notes.**

bra, *Elatine brachysperma*; hyd, *E. hydropiper*; tri, *E. triandra*; ort, *E. orthosperma*; hex, *E. hexandra*; mac, *E. macropoda*; gus, *E. gussonei*; cal, *E. californica*; hun, *E. hungarica*; T, Terrestrial; A, Aquatic.

**Table 3** Confusion matrix from Random Forest classification based on vegetative traits.

| | braAq | braTe | calAq | calTe | gusAq | gusTe | hexAq | hexTe | hunAq | hunTe | hydAq | hydTe | macAq | macTe | ortAq | ortTe | triAq | triTe | Classification error |
|---|---|---|---|---|---|---|---|---|---|---|---|---|---|---|---|---|---|---|---|
| braAq | 38 | 0 | 0 | 1 | 0 | 0 | 3 | 3 | 0 | 2 | 0 | 0 | 0 | 0 | 0 | 3 | 0 | 0 | 0.24 |
| braTe | 0 | 43 | 1 | 1 | 0 | 0 | 2 | 1 | 0 | 0 | 1 | 0 | 0 | 0 | 1 | 0 | 0 | 0 | 0.14 |
| calAq | 0 | 0 | 37 | 7 | 0 | 0 | 0 | 0 | 0 | 0 | 0 | 1 | 1 | 0 | 4 | 0 | 0 | 0 | 0.26 |
| calTe | 0 | 0 | 6 | 40 | 0 | 0 | 0 | 1 | 0 | 0 | 0 | 3 | 0 | 0 | 0 | 0 | 0 | 0 | 0.20 |
| gusAq | 0 | 0 | 0 | 0 | 47 | 2 | 0 | 0 | 0 | 0 | 0 | 0 | 0 | 1 | 0 | 0 | 0 | 0 | 0.06 |
| gusTe | 0 | 0 | 1 | 0 | 2 | 47 | 0 | 0 | 0 | 0 | 0 | 0 | 0 | 0 | 0 | 0 | 0 | 0 | 0.06 |
| hexAq | 0 | 5 | 0 | 1 | 0 | 0 | 27 | 2 | 0 | 3 | 3 | 2 | 0 | 0 | 0 | 0 | 2 | 5 | 0.46 |
| hexTe | 4 | 0 | 0 | 1 | 0 | 0 | 0 | 33 | 0 | 3 | 0 | 4 | 0 | 0 | 1 | 3 | 0 | 1 | 0.34 |
| hunAq | 0 | 0 | 0 | 0 | 0 | 0 | 0 | 0 | 50 | 0 | 0 | 0 | 0 | 0 | 0 | 0 | 0 | 0 | 0.00 |
| hunTe | 0 | 0 | 0 | 2 | 0 | 0 | 0 | 1 | 0 | 31 | 0 | 9 | 0 | 0 | 1 | 5 | 0 | 1 | 0.38 |
| hydAq | 0 | 3 | 0 | 0 | 0 | 0 | 0 | 0 | 0 | 0 | 43 | 0 | 0 | 0 | 3 | 0 | 0 | 1 | 0.14 |
| hydTe | 0 | 2 | 0 | 3 | 0 | 0 | 2 | 2 | 0 | 11 | 0 | 28 | 0 | 0 | 2 | 0 | 0 | 0 | 0.44 |
| macAq | 0 | 0 | 0 | 0 | 1 | 0 | 0 | 0 | 0 | 0 | 0 | 0 | 48 | 1 | 0 | 0 | 0 | 0 | 0.04 |
| macTe | 0 | 0 | 2 | 0 | 1 | 0 | 0 | 0 | 0 | 0 | 0 | 0 | 1 | 46 | 0 | 0 | 0 | 0 | 0.08 |
| ortAq | 0 | 0 | 1 | 0 | 0 | 0 | 0 | 0 | 0 | 0 | 0 | 3 | 0 | 0 | 46 | 0 | 0 | 0 | 0.08 |
| ortTe | 3 | 0 | 0 | 0 | 0 | 0 | 0 | 2 | 0 | 1 | 0 | 1 | 0 | 0 | 0 | 43 | 0 | 0 | 0.14 |
| triAq | 0 | 2 | 0 | 0 | 0 | 0 | 1 | 0 | 0 | 0 | 0 | 0 | 0 | 0 | 0 | 0 | 47 | 0 | 0.06 |
| triTe | 0 | 0 | 0 | 0 | 0 | 0 | 3 | 0 | 0 | 0 | 0 | 0 | 0 | 0 | 0 | 0 | 0 | 47 | 0.06 |

**Notes.**

Abbreviations as in Table 2.

**Table 4 Kruskal–Wallis groups based on vegetative characters. The significance level set to 0.05.** Unique letters indicate significance different groups while the same letters mean statistically not different subsets.

| | Stem | Pedicel | 1st petiole | 1st leaf |
|---|---|---|---|---|
| braAq | a | a | a | a |
| braTe | bcd | a | bc | bcd |
| calAq | bef | ab | b | efgh |
| calTe | cd | bc | cde | efg |
| gusAq | eg | de | bcd | ae |
| gusTe | g | d | de | efgh |
| hexAq | cdh | c | bcd | bci |
| hexTe | ah | c | cde | fgh |
| hunAq | ch | ce | f | bd |
| hunTe | a | c | bcd | efg |
| hydAq | bdf | a | f | bd |
| hydTe | ah | a | bcd | fhi |
| macAq | bef | d | f | a |
| macTe | efg | de | f | aeg |
| ortAq | bcd | a | f | chi |
| ortTe | a | a | ae | a |
| triAq | bcd | a | b | d |
| triTe | a | a | de | bcd |

**Notes.**
Abbreviations as in Table 2.

on shared vegetative characteristics. Additionally, *Elatine gussonei*, which is an enigmatic plant of the Mediterranean was firstly described as a variety of *Elatine hydropiper* and was later classified as a separate species based on the shape of the seed and the length of flowers pedicels (*Brullo et al., 1988*).

The results and method applied in this study provide much evidence to explain why seed traits are better than vegetative traits in taxonomy of *Elatine*. Although some students of the genus were arguing for the taxonomic importance of pedicel length (*Seubert, 1845*; *Moesz, 1908*; *Cook, 1968a*), others had expressed doubts about its relevance, and clearly attributed morphological variation to response to environmental differences (*Margittai, 1939*; *Soó, 1974*). Our results indicate that vegetative characters have less taxonomic relevant information than what was usually considered before. It suggests that it is not appropriate to use vegetative traits in species identification within the genus *Elatine*.

The investigation of the extent of phenotypic plasticity of seed and vegetative traits in nine *Elatine* species grown in different environmental circumstances gave a clear answer to the above debate. Although only one clone of each field-collected specimen was investigated, this assured that the reported difference between the different ecotypes of the same clone stands for phenotypic plasticity and it is not influenced by genotypic difference. The similar placement of different ecotypes of the same species in the seed trait based multivariate analyses (Fig. 3) indicates clearly the stability of seed characters. Secondly, we consider this relatively limited sampling to be still the most comprehensive

**Table 5** **Cross validated classification based on seed traits.** Rows: given group; columns: predicted groups. Only 50.7% of the specimens are correctly assigned.

| | braTe | braAq | calTe | calAq | gusTe | gusAq | hexTe | hexAq | hunTe | hunAq | hydTe | hydAq | macTe | macAq | ortTe | ortAq | triTe | triAq | Total |
|---|---|---|---|---|---|---|---|---|---|---|---|---|---|---|---|---|---|---|---|
| braTe | 10 | 29 | 0 | 0 | 0 | 0 | 0 | 0 | 0 | 0 | 0 | 0 | 1 | 0 | 0 | 0 | 7 | 3 | 50 |
| braAq | 9 | 41 | 0 | 0 | 0 | 0 | 0 | 0 | 0 | 0 | 0 | 0 | 0 | 0 | 0 | 0 | 0 | 0 | 50 |
| calTe | 0 | 0 | 15 | 16 | 0 | 0 | 0 | 0 | 11 | 8 | 0 | 0 | 0 | 0 | 0 | 0 | 0 | 0 | 50 |
| calAq | 0 | 0 | 16 | 14 | 0 | 0 | 0 | 0 | 12 | 8 | 0 | 0 | 0 | 0 | 0 | 0 | 0 | 0 | 50 |
| gusTe | 0 | 0 | 0 | 0 | 38 | 10 | 0 | 0 | 1 | 0 | 0 | 0 | 1 | 0 | 0 | 0 | 0 | 0 | 50 |
| gusAq | 0 | 0 | 0 | 0 | 14 | 30 | 0 | 0 | 2 | 2 | 0 | 0 | 2 | 0 | 0 | 0 | 0 | 0 | 50 |
| hexTe | 0 | 0 | 0 | 0 | 0 | 0 | 24 | 21 | 0 | 0 | 0 | 0 | 1 | 4 | 0 | 0 | 0 | 0 | 50 |
| hexAq | 0 | 0 | 0 | 0 | 0 | 0 | 19 | 26 | 0 | 1 | 0 | 0 | 3 | 1 | 0 | 0 | 0 | 0 | 50 |
| hunTe | 0 | 0 | 10 | 6 | 0 | 7 | 0 | 0 | 23 | 3 | 0 | 0 | 1 | 0 | 0 | 0 | 0 | 0 | 50 |
| hunAq | 0 | 0 | 8 | 3 | 7 | 3 | 0 | 0 | 6 | 23 | 0 | 0 | 0 | 0 | 0 | 0 | 0 | 0 | 50 |
| hydTe | 0 | 0 | 2 | 0 | 0 | 0 | 0 | 0 | 0 | 0 | 31 | 17 | 0 | 0 | 0 | 0 | 0 | 0 | 50 |
| hydAq | 0 | 0 | 1 | 0 | 0 | 0 | 0 | 0 | 0 | 0 | 16 | 33 | 0 | 0 | 0 | 0 | 0 | 0 | 50 |
| macTe | 0 | 0 | 0 | 0 | 0 | 0 | 3 | 2 | 0 | 0 | 0 | 0 | 29 | 16 | 0 | 0 | 0 | 0 | 50 |
| macAq | 0 | 0 | 0 | 0 | 0 | 0 | 7 | 7 | 0 | 0 | 0 | 0 | 22 | 13 | 0 | 0 | 1 | 0 | 50 |
| ortTe | 0 | 0 | 0 | 0 | 0 | 0 | 0 | 0 | 0 | 0 | 0 | 0 | 0 | 0 | 29 | 21 | 0 | 0 | 50 |
| ortAq | 0 | 0 | 0 | 0 | 0 | 0 | 0 | 2 | 0 | 0 | 0 | 0 | 0 | 0 | 26 | 22 | 0 | 0 | 50 |
| triTe | 7 | 0 | 0 | 0 | 0 | 0 | 0 | 0 | 0 | 0 | 0 | 0 | 2 | 0 | 0 | 0 | 23 | 18 | 50 |
| triAq | 0 | 0 | 0 | 0 | 0 | 0 | 0 | 0 | 0 | 0 | 0 | 0 | 0 | 0 | 0 | 0 | 22 | 28 | 50 |
| Total | 26 | 70 | 52 | 39 | 59 | 50 | 53 | 58 | 55 | 45 | 47 | 50 | 62 | 34 | 55 | 43 | 53 | 49 | 900 |

**Notes.**

Abbreviations as in Table 2.

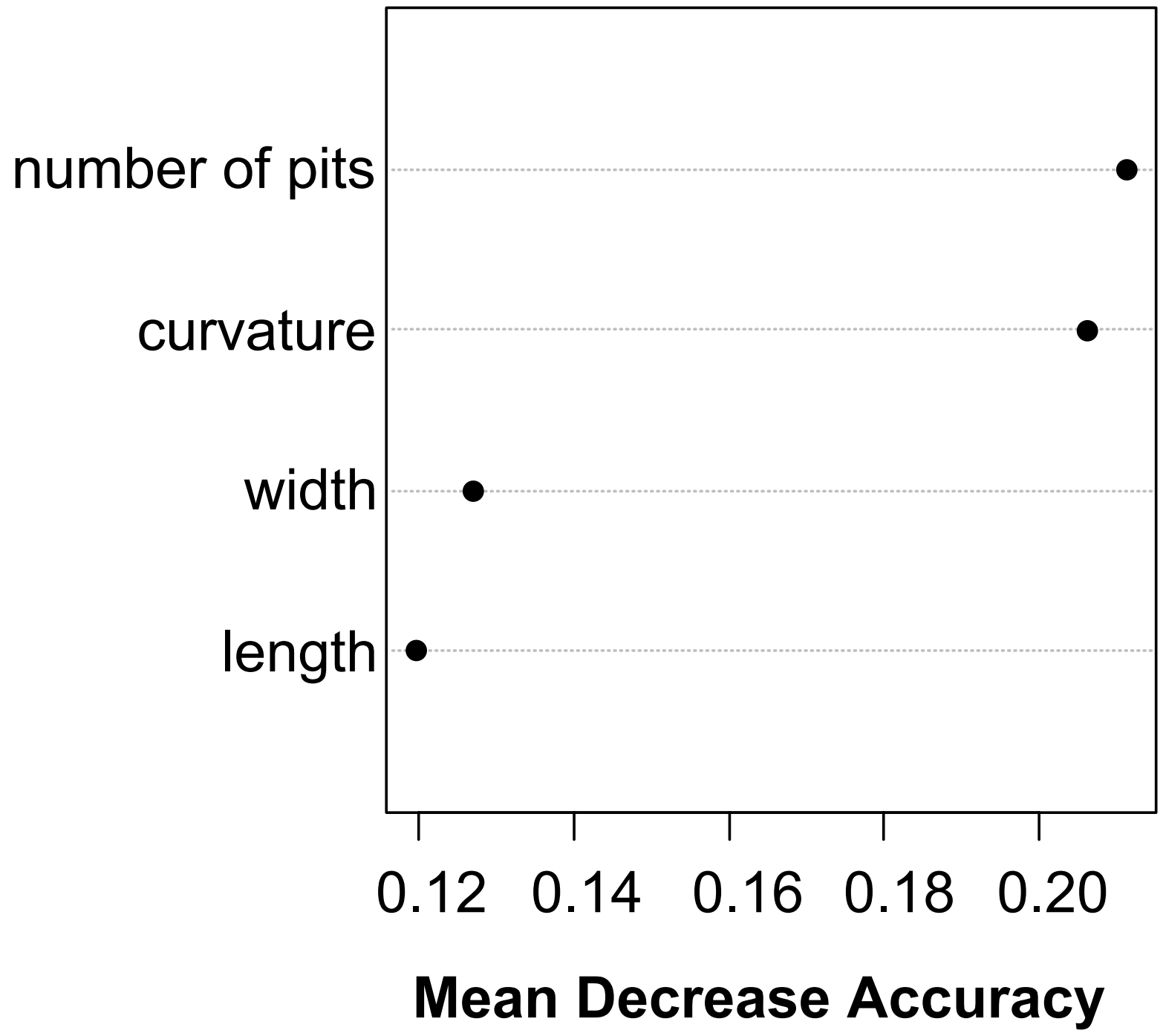

**Figure 6** **Dotchart of variable importance as measured by a Random Forest for seed traits.**

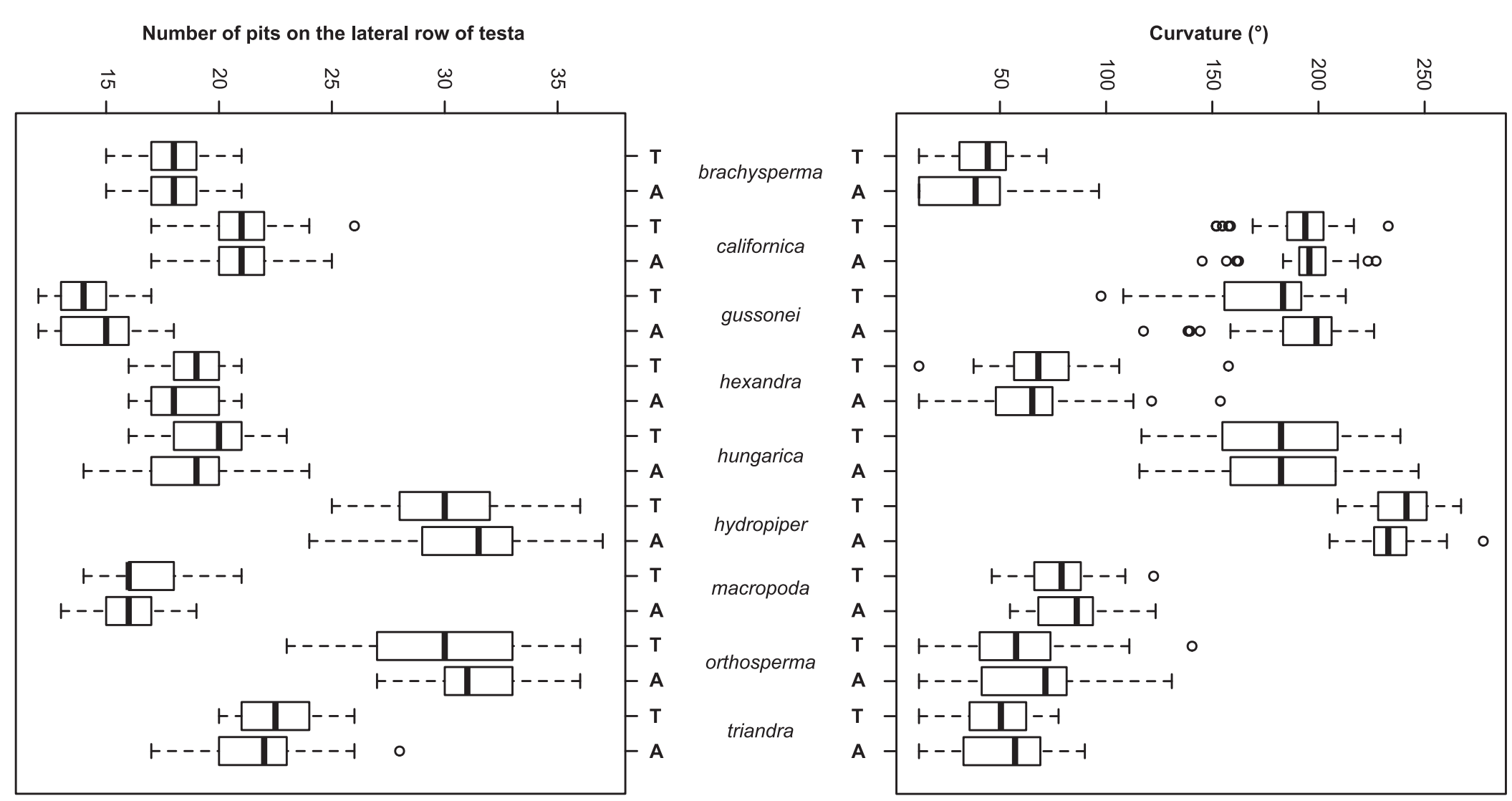

**Figure 7** **Boxplots of the most discriminative seed traits among the nine *Elatine* species studied. Terrestrial (T) and aquatic (A) forms are not significantly different in all the species.** Notations: Boxes mean 25–75 percentiles, lines are medians, squares are mean, whiskers are standard deviations.

**Table 6** **Confusion matrix from Random Forest classification based on seed traits.**

| | braAq | braTe | calAq | calTe | gusAq | gusTe | hexAq | hexTe | hunAq | hunTe | hydAq | hydTe | macAq | macTe | ortAq | ortTe | triAq | triTe | Class. error |
|---|---|---|---|---|---|---|---|---|---|---|---|---|---|---|---|---|---|---|---|
| braAq | 29 | 21 | 0 | 0 | 0 | 0 | 0 | 0 | 0 | 0 | 0 | 0 | 0 | 0 | 0 | 0 | 0 | 0 | 0.42 |
| braTe | 26 | 19 | 0 | 0 | 0 | 0 | 0 | 0 | 0 | 0 | 0 | 0 | 1 | 0 | 0 | 0 | 2 | 2 | 0.62 |
| calAq | 0 | 0 | 18 | 19 | 0 | 0 | 0 | 0 | 4 | 8 | 0 | 1 | 0 | 0 | 0 | 0 | 0 | 0 | 0.64 |
| calTe | 0 | 0 | 20 | 20 | 0 | 0 | 0 | 0 | 3 | 7 | 0 | 0 | 0 | 0 | 0 | 0 | 0 | 0 | 0.60 |
| gusAq | 0 | 0 | 0 | 0 | 34 | 13 | 0 | 0 | 0 | 1 | 0 | 0 | 0 | 2 | 0 | 0 | 0 | 0 | 0.32 |
| gusTe | 0 | 0 | 0 | 0 | 16 | 32 | 0 | 0 | 1 | 1 | 0 | 0 | 0 | 0 | 0 | 0 | 0 | 0 | 0.36 |
| hexAq | 0 | 0 | 0 | 0 | 0 | 0 | 18 | 21 | 1 | 0 | 0 | 0 | 7 | 2 | 0 | 0 | 0 | 1 | 0.64 |
| hexTe | 0 | 0 | 0 | 0 | 0 | 0 | 19 | 24 | 1 | 0 | 0 | 0 | 5 | 1 | 0 | 0 | 0 | 0 | 0.52 |
| hunAq | 0 | 0 | 3 | 5 | 3 | 2 | 0 | 0 | 25 | 12 | 0 | 0 | 0 | 0 | 0 | 0 | 0 | 0 | 0.50 |
| hunTe | 0 | 0 | 8 | 7 | 4 | 2 | 0 | 0 | 6 | 23 | 0 | 0 | 0 | 0 | 0 | 0 | 0 | 0 | 0.54 |
| hydAq | 0 | 0 | 0 | 0 | 0 | 0 | 0 | 0 | 0 | 0 | 37 | 13 | 0 | 0 | 0 | 0 | 0 | 0 | 0.26 |
| hydTe | 0 | 0 | 0 | 0 | 0 | 0 | 0 | 0 | 0 | 0 | 20 | 30 | 0 | 0 | 0 | 0 | 0 | 0 | 0.40 |
| macAq | 0 | 1 | 0 | 0 | 0 | 0 | 5 | 3 | 0 | 0 | 0 | 0 | 20 | 21 | 0 | 0 | 0 | 0 | 0.60 |
| macTe | 0 | 0 | 0 | 0 | 1 | 0 | 4 | 1 | 0 | 0 | 0 | 0 | 24 | 20 | 0 | 0 | 0 | 0 | 0.60 |
| ortAq | 0 | 0 | 0 | 0 | 0 | 0 | 0 | 0 | 0 | 0 | 0 | 0 | 0 | 0 | 23 | 27 | 0 | 0 | 0.54 |
| ortTe | 0 | 0 | 0 | 0 | 0 | 0 | 0 | 0 | 0 | 0 | 0 | 0 | 0 | 0 | 25 | 25 | 0 | 0 | 0.50 |
| triAq | 0 | 1 | 0 | 0 | 0 | 0 | 0 | 0 | 0 | 0 | 0 | 0 | 0 | 0 | 1 | 0 | 26 | 22 | 0.48 |
| triTe | 1 | 2 | 0 | 0 | 0 | 0 | 1 | 0 | 0 | 0 | 0 | 0 | 0 | 1 | 1 | 0 | 27 | 17 | 0.66 |

**Notes.**

Abbreviations as in Table 2.

**Table 7 Kruskal–Wallis groups based on seed traits. The significance level set to 0.05.** Unique letters indicates significantly different groups while the same letters mean significantly not different subsets.

| ID | Number of pits | Curvature |
|---|---|---|
| braAq | abc | ab |
| braATe | abcd | a |
| calAq | efg | cde |
| calTe | defg | cde |
| gusAq | h | c |
| gusTe | h | cde |
| gusAq | abcde | bf |
| hexTe | abc | abf |
| hunAq | adef | c |
| hunTe | abde | cd |
| hydAq | ij | e |
| hydTe | i | de |
| macAq | bch | f |
| macTe | ch | f |
| ortAq | ij | abf |
| ortTe | i | abf |
| triAq | gj | ab |
| triTe | fg | abf |

**Notes.**
Abbreviations as in Table 2.

experimental study in the genus, thus we regard our data and conclusions as pioneering in the genus.

Different species boundaries were indicated by the statistical analysis of different set of vegetative and seed traits. On one hand our results clearly demonstrate that aquatic or terrestrial conditions can induce morphological alteration (i.e., different appearance of the same species), thus, we can conclude that vegetative traits are highly influenced by environmental factors. Moreover, we found various morphological distances between the different ecological forms of the same species according to vegetative traits. The morphological distance between the different ecological forms showed a large heterogeneity and nearly all was statistically significant. For example the aquatic and terrestrial forms of *E. macropoda*, *E. californica* and *E. gussonei* were only slightly different and the two forms clustered to the same branch in the UPGMA tree, whereas the morphological distance between the two forms of *E. triandra* is bigger than the difference between species. Because of the previously described instability, the vegetative trait based identification is not reliable and could lead to erroneous species identification. Consequently, the usage of vegetative traits in some literature sources (e.g., *Moesz, 1908*) to separate species needs careful re-evaluation and highly cautious use. In fact the total ignorance of phenotypic plasticity in *Elatine* taxonomy might lead to much narrower species concepts then would be necessary to apply in such a genus. An example can be the report of *E. ambigua* from Europe (*Moesz, 1908*). We suspect this plant was a form of *E. triandra* with elongated pedicels, what is otherwise the

distinguishing character between the two species. If a more wide species concept had been applied, the specimen could have been correctly identified as *E. triandra*.

Vegetative and generative traits are affected by different selection forces (*Grime, 2001*). Vegetative organs play an important role in photosynthesis and the physical maintenance of the whole plant in various and often changing environments. Phenotypic plasticity (i.e., the morphological alteration of plants vegetative organs) is the most important adaptation of plants to temporal and spatial environmental variability (*Sultan, 2000*). Plasticity gives opportunities for plants to improve their resource acquisition, resistance, and adaptability to stress and disturbance (*Grime, Crick & Rincon, 1986*). The significant vegetative variability of the amphibious genus *Elatine* therefore plays a key role in adaptation to starkly different environmental conditions. Seed traits belong to generative traits with the basic role of propagation, and could similarly vary under different habitat characteristics (i.e., aquatic or terrestrial). Nevertheless, we found seed traits to be more stable. Although different environmental conditions can influence some reproductive traits in aquatic plants, but this phenomenon recognized only in seed numbers (*Garbey, Thiébaut & Muller, 2004*), seed mass (*Fenner & Thompson, 2005*) and seed size (*Westoby, Jurado & Leishmann, 1992*), and not in seed morphology. Most probably reproductive traits are under a selective pressure that favors stability even in different habitat characteristics. Disregarding the reason behind the stability of seed traits in the amphibious genus *Elatine*—similar to other plant species—reproductive characteristics are favorable in species identification.

Based on our analyses seed characters of aquatic and terrestrial forms of the same species were not statistically different from each other, except in few cases, when we suspect phylogenetically independent occurrence of the same character. Contrary to our findings based on the vegetative traits, the morphological distance between seeds of two ecological forms of the same species were very small as seen on the UPGMA tree (Fig. 3C). Thus, seed traits show more stability under different environmental influence than vegetative traits. Among the measured seed traits the curvature and the number of pits had the biggest standardised loadings on the first and the second discriminant function, thus proved to be useful for identifying species. Based on seed characteristics, all European species form distinct groups. There is only one species pair where the separation is not possible based on seed traits: the Eurasian *E. hungarica* and North-American *E. californica*, which have similar seeds. Whether this shared morphology is due to phylogenetic relatedness or simple morphological homoplasy warrants for further research.

## ACKNOWLEDGEMENTS

We are very grateful to Csaba Máthé, Márta M-Hamvas, Pertti Uotila and Luis Eguiarte for their professional comments and linguistic improvements on the earlier draft of this paper. The authors would like to thank Viktor Löki (Debrecen) for collecting *Elatine californica* and *E. brachysperma* samples in the US, Zsanett Ajtay and Anna Farkas (Debrecen) for their assistance during laboratory work, and Nikola Rahmé (Budapest) for photographing seeds.

## Funding

This research was supported by the European Union and the State of Hungary, co-financed by the European Social Fund in the framework of TÁMOP-4.2.4.A/2-11/1-2012-0001 'National Excellence Program' and TÁMOP-4.2.2B-15/1/KONV-2015-0001 program. Instrumental and infrastructural supports of the OTKA K108992 and OTKA PD109686 Grants and National Science Center (Poland) N N303 470638 Grant (AP) are also highly appreciated, and was a Bolyai Fellowship (GS) from the Hungarian Academy of Sciences. The funders had no role in study design, data collection and analysis, decision to publish, or preparation of the manuscript.

## Grant Disclosures

The following grant information was disclosed by the authors:
European Social Fund: TÁMOP-4.2.4.A/2-11/1-2012-0001, TÁMOP-4.2.2B-15/1/KONV-2015-0001.
OTKA: K108992, PD109686.
National Science Center (Poland): N N303 470638.
Hungarian Academy of Sciences: Bolyai Fellowship (GS).

## Competing Interests

The authors declare there are no competing interests.

## Author Contributions

- Attila Molnár V. conceived and designed the experiments, performed the experiments, contributed reagents/materials/analysis tools, wrote the paper, prepared figures and/or tables, reviewed drafts of the paper.
- János Pál Tóth conceived and designed the experiments, analyzed the data, wrote the paper, prepared figures and/or tables, reviewed drafts of the paper.
- Gábor Sramkó wrote the paper, reviewed drafts of the paper.
- Orsolya Horváth conceived and designed the experiments, performed the experiments, contributed reagents/materials/analysis tools, reviewed drafts of the paper.
- Agnieszka Popiela and Attila Mesterházy contributed reagents/materials/analysis tools, reviewed drafts of the paper.
- Balázs András Lukács contributed reagents/materials/analysis tools, wrote the paper, reviewed drafts of the paper.

## Field Study Permissions

The following information was supplied relating to field study approvals (i.e., approving body and any reference numbers):

*Elatine hungarica*, *E. hydropiper* and *E. triandra* are protected species and were sampled in Hungary with the permission of the Hortobágy National Park Directorate (Permission id.: 45-2/2000, 250-2/2001).

## Data Availability

Raw data were uploaded as Supplemental Information.

## Supplemental Information

Supplemental information for this article can be found online at http://dx.doi.org/10.7717/peerj.1473#supplemental-information.

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
