# Peer review of "Flood induced phenotypic plasticity in amphibious genus Elatine (Elatinaceae)"

_PeerJ, doi:10.7717/peerj.1473_

## Round 0.1 · original submission · Major Revisions

This is an interesting study of a poorly know group of angiosperms. Apparently this poor knowledge is at least in part the result of problems to define and characterize the species, due to the plasticity of the vegetative morphology. This may be especially important in the case of species in danger of extinction.

Both reviewers are deeply concerned about the use of a single genotype per species. While I think the sampling may be adequate in showing that there is an overlap among species in the measured vegetative traits, the authors need to do a better job in justifying this decision.

I also would like a better discussion on other not-measured reproductive trait, in particular flowers: How variable (an useful for taxonomy) are flowers and fruits among species?

Please, answer all the concerns of the reviewers in the next version of the manuscript.
One reviewer had some problems with the spell checker in his computer (i.e., an “species” meant “specific”), but it is possible to understand all the comments.

Reviewer 1 ·

Basic reporting

In this study authors analyzed vegetative and reproductive traits of nine species under terrestrial and aquatic conditions to clarify whether vegetative or reproductive traits are good indicators of taxonomic differences among species. The study has a serious problem with the experimental design (see below) given that only one genotype was sampled per species. Besides this problem, analyses and results.The two commnets below exemplified two of my major concerns around this study.

Line 95-101: Please try to be more species here. Which traits are now being used? I suspect that what the authors measured were those that are used plus a complement of other potentially useful traits. Vegetative traits used in this study are quite labile under natural conditions. It looks like they should not be used for species identification. I would show the reader a complete list of vegetative and reproductive traits used for taxonomic purposes in the genus. As it is presented reproductive traits are already used in the taxonomy of the genus. Be more specific about the problem of using vegetative traits.

Line 108-113: The problem is not whether vegetative traits are plastic, which indeed they are, but whether species can be taxonomically identified using vegetative traits vs reproductive traits. In most plant species reproductive traits are more stable and one would expect the same in this case. To me a more direct examination of the authors concern would have been the analysis of the phenotypic overlap among species in vegetative and reproductive traits. If vegetative traits used for taxonomy do not overlap, despite being plastic, then they could be used to differentiate among species. Results provide consistent evidence of little or no overlap between species for reproductive trait, and confidence intervals also support this pattern, but the formulation of the main goals can be re-oriented to answer this question in a more compelling way focusing on variability as well as on mean values.

Experimental design

Line 121-124: OK but this sampling criteria can bias the estimated similarity/differences among species both in vegetative and reproductive traits. I would expect that natural genetic variation was sampled to obtain a representative estimate of mean values for each species. Moreover, if all species are highly autogamous, clones may show larger differences among them reducing differences among species. These two sources of variation should have been estimated and neither were replicated in the present study. Taking only one genotype per species can increase the probability of detecting differences in the analyses.

Validity of the findings

The study provide convincing evidence that reprodutive (seed) traits are more stable than vegetative traits.

Additional comments

Independently of whether someone mentioned 60 years ago that a rigorous examination of phenotypic vegetative and reproductive traits should be performed in these set of species, genetic data should be included to support species differentiation criteria based on vegetative or reproductive traits. The expectation that vegetative traits are more plastic and variable than reproductive traits is quite obvious given the tremendous amount of empirical data that support such pattern. The argumentation provided in the Discussion that some of these species have a high conservation status is rather relevant and can be used in the Introduction to support the need of further analyses of vegetative and reproductive trait variation for taxonomic purposes.

Reviewer 2 ·

Basic reporting

The manuscript reports an experimental study to characterize the expression of phenotypic plasticity of vegetative and seed traits in aquatic and terrestrial forms of nine Elatine species and explore which traits of both forms for the same species are suitable to taxonomic identification. Using several multivariate statistical analysis, as well as Kruskal-Wallis tests, the authors found that seed traits were more consistent between aquatic and terrestrial forms of the same species. The questions and statements are very clear. The manuscript is well written but I have several comments and suggestion that should be considered by the authors.

Experimental design

There are some details of the experiment that the authors should provide:

Lines 130-131: How long the plants were exposed to both environments? How long duration of the experiment?

Lines 131-133: Six morphological traits? There are more than six morphological traits included in the analysis (for instance see Figure 6).

Lines 133-134: Each specimen reached 50 fruiting stems? Each "fruiting stem" was considered as a clone?

Lines 134-135: From these specimens, the seeds were collected? Just one seed per clone? How the authors did this? Please clarify.

Lines 168-169: I do not believe that Kruskal-Wallis tests are necessary. The multivariate analyses appear to be more powerful.

Validity of the findings

The main result of the study is that seed traits are more stable between aquatic and terrestrial forms; consequently such traits are suitable to distinguish among species. However, there are two issues that the authors must include in the Discussion:

1) Lines 258-270: The use of only one genotype by species constitutes a solid and reliable evidence to support the conclusions reached in this study? I think that the authors could justify better the use of only one clone (genotype) of each species in the study. In addition, the authors point out that "... the similar placement of samples from different populations of the same species...” when they collected seeds of a single population of each species.

2) The authors should argue and discuss why the seed morphology is more stable between aquatic and terrestrial forms.

Additional comments

1) Lines 150-152: This is a long phrase and it is not clear for me.

2) Line 167: must be “matrix”.

3) Lines 185-186: “The cross validated classification correctly assigned 77.7% of the specimens” (as appears in the text) or “The 82.8 % of the specimens are correctly assigned” (as appears in Table 3)

4) I think that Tables 2 and 6 are unnecessary. The description of the results of the NPMANOVAs in the lines 191-191 (vegetative traits) and 219-223 (seed traits) are enough.

---

## Round 0.2 · Minor Revisions

The manuscript clearly improved after answering the concerns and questions of the reviewers. Nevertheless, some issues need to be addressed before the paper can be accepted.

The main issue is that Figures 4 and 6 are not mentioned in the text. The figures need to be indicated and explained in the Results, or removed if considered unnecessary in this new version (but I think they are useful and informative).

Also, Figure 7 is cited in line 219 while Fig. 5 is mentioned for the first time in line 227. Rename the figures so they appear in a consecutive way.

Please, carefully review the next version the manuscript, in particular the Figures, Tables, their captions and heads, and the references and other editorial details.

Minor comments:
Abstract:
Line 36: “50-50” is a confusing way to state the sample sizes. I suggest something like “were measured on 50 fruiting stems of the aquatic and on 50 stems of the terrestrial form of the same clone”.

Introduction:
Line 90: Improve this line. Something like: “because large clonal plants are especially exposed to variation in water…”.
Line 93: Uotila, 2009a, 2010. Remove the “a”, as there is only one paper by this author in that year, and the 2010 reference is missing.
Lines 134: Please also state the time it took to reach the 50 stems, “between xx and yyy days”.
Lines 136 and 139: As in the Abstract, change the “50-50” for a clearer explanation..
Lines 176: Remove one “.”

Results:
Lines 180- to 183: Improve the writing, consider something like:
“The vegetative traits of the aquatic or terrestrial forms of the nine Elatine species were different with high discriminatory power (Wilks’s λ= 0.0001, p<0.001). “
Lines 204-205: Also improve these lines as suggested above.

Discussion:
Lines 233 to 234: Remove “but mostly in response to different range of environmental factors.” As this is clear from the line 231.
Lines 251 to 254: You have to mention your conclusion regarding the relevance of the pedicel length from you analysis.
Lines 257 to 261: Remove red color.
Lines 284 to 299: Remove red color.
Line 294: A word is missing between “this” and “recognized”.

References:
Lines 404 and 406: Indicate which reference is Molnár a, or b, as cited in the text.
Line 419: Remove parenthesis from year.

Figures:
Please check all Figure captions, as in the PDF many words are not separated.
Figure 5: Improve figure caption: “Box plots of the four vegetative traits studied among the nine Elatine species studied.” Remove the first “studied” and mention which four, since a total 6 traits were studied.
Figure 6: Improve the caption, I suggest: “Dot chart of variable importance for vegetative traits as measured by a Random Forest (reference).”

Table 1: Change “Popiela et al. accepted” to “Popiela et al. 2015) as is in the references.

=====

---

## Round 0.3 · Minor Revisions

The manuscript is almost ready, but there are three sections were it still needs some improvement.

I am attaching a pdf file from an annotated Word document with my suggestions.

Following the line number of the Word documents, these are my three concerns.

Line 215: Change “have” for “has”.

Line 256: I suggest changing it to “seed traits did not differ between”.

Lines 298 and 299: This phrase needs to be improved. I suggest something like (but you have to see if this is want you wanted to say): “Our results indicate that vegetative characters have less taxonomic relevant information than what was usually considered before.”

---

## Round 0.4 · accepted · Accept

I consider that this is an interesting and relevant study of a poorly known group of angiosperms.

This new revision answers all my concerns. The study will be useful for the future understanding of the group, and I am glad to accept this new version of the paper.

I appreciate and thank your efforts to answer all the questions and criticism by me and by the reviewers.